# Exploring the Genomic Diversity and Antimicrobial Susceptibility of *Bifidobacterium pseudocatenulatum* in a Vietnamese Population

Hao Chung The,[a] Chau Nguyen Ngoc Minh,[a] Chau Tran Thi Hong,[a] To Nguyen Thi Nguyen,[a] Lindsay J. Pike,[b] Caroline Zellmer,[c,d] Trung Pham Duc,[a] Tuan-Anh Tran,[a] Tuyen Ha Thanh,[a] Minh Pham Van,[a] Guy E. Thwaites,[a,e] Maia A. Rabaa,[a,e] Lindsay J. Hall,[f,g,h] Stephen Baker[b,c]

[a]Oxford University Clinical Research Unit, Ho Chi Minh City, Vietnam
[b]The Wellcome Sanger Institute, Hinxton, Cambridge, United Kingdom
[c]University of Cambridge School of Clinical Medicine, Cambridge Biomedical Campus, Cambridge, United Kingdom
[d]Department of Medicine, University of Cambridge School of Clinical Medicine, Cambridge Biomedical Campus, Cambridge, United Kingdom
[e]Centre for Tropical Medicine and Global Health, Nuffield Department of Clinical Medicine, Oxford University, Oxford, United Kingdom
[f]Quadram Institute Biosciences, Norwich, United Kingdom
[g]Norwich Medical School, University of East Anglia, Norwich, United Kingdom
[h]Intestinal Microbiome, School of Life Sciences, ZIEL - Institute for Food & Health, Technical University of Munich, Freising, Germany

Hao Chung The and Chau Nguyen Ngoc Minh contributed equally to this article. The author order was determined by their equal but gradated contributions for this paper.

**ABSTRACT** *Bifidobacterium pseudocatenulatum* is a member of the human gut microbiota, and specific variants of *B. pseudocatenulatum* have been associated with health benefits such as improving gut integrity and reducing inflammatory responses. Here, we aimed to assess the genomic diversity and predicted metabolic profiles of *B. pseudocatenulatum* cells found colonizing the gut of healthy Vietnamese adults and children. We found that the population of *B. pseudocatenulatum* from each individual was distinct and highly diverse, with intraclonal variation attributed largely to a gain or loss of carbohydrate-utilizing enzymes. The *B. pseudocatenulatum* genomes were enriched with glycosyl hydrolases predicted to target plant-based nondigestible carbohydrates (GH13, GH43) but not host-derived glycans. Notably, the exopolysaccharide biosynthesis region from organisms isolated from healthy children showed extensive genetic diversity and was subject to a high degree of genetic modification. Antimicrobial susceptibility profiling revealed that the Vietnamese *B. pseudocatenulatum* cells were uniformly susceptible to beta-lactams but exhibited variable resistance to azithromycin, tetracycline, ciprofloxacin, and metronidazole. The genomic presence of *ermX* and *tet* variants conferred resistance against azithromycin and tetracycline, respectively; ciprofloxacin resistance was associated with a mutation(s) in the quinolone resistance-determining region (GyrA, S115, and/or D119). Our work provides the first detailed genomic and antimicrobial resistance characterization of *B. pseudocatenulatum* found in the Vietnamese population, which can be exploited for the rational design of probiotics.

**IMPORTANCE** *Bifidobacterium pseudocatenulatum* is a beneficial member of the human gut microbiota. The organism can modulate inflammation and has probiotic potential, but its characteristics are largely strain dependent and associated with distinct genomic and biochemical features. Population-specific beneficial microbes represent a promising avenue for the development of potential probiotics, as they may exhibit a more suitable profile in the target population. This study investigates the underexplored diversity of *B. pseudocatenulatum* in Vietnam and provides more understanding of its genomic diversity, metabolic potential, and antimicrobial susceptibility. Such data from indigenous populations are essential for selecting probiotic candidates that can be accelerated into further preclinical and clinical investigations.

Address correspondence to Hao Chung The, haoct@oucru.org.

We studied the genomic diversity and AMR of the under-explored probiotic organism *Bifidobacterium pseudocatenulatum* in the Vietnamese population.

KEYWORDS *Bifidobacterium* antimicrobial resistance, *Bifidobacterium* probiotic, *Bifidobacterium* genome, *Bifidobacterium pseudocatenulatum*, antimicrobial resistance, *Bifidobacteria*, developing country, exopolysaccharide, genomic diversity, glycosyl hydrolase

*B*ifidobacterium is a genus of Gram-positive non-spore-forming anaerobic bacteria and among the most well-studied members of the human gut microbiota (1). These bacteria are a major component of the gut microbiota and are transferred vertically from mothers to newborns (2). They are also measurably enriched in babies delivered vaginally compared to those delivered via caesarean section (3). Several health-promoting benefits are associated with *Bifidobacterium* colonization of the human gut. These benefits are associated with the production of primary and secondary metabolites, immunomodulatory activities, and protection from infections (1, 4–6). The genus is composed of multiple human-adapted species, many of which colonize the gut during different life stages; this colonization pattern is largely dependent on the dominant carbohydrate sources available in the intestinal lumen (7). The saccharolytic lifestyle of *Bifidobacterium* can be observed by its ability to catabolize a wide variety of carbohydrates (from monosaccharides to complex plant-derived polysaccharides), which are ultimately channeled into a unique hexose metabolic pathway ("bifid shunt") (8). Certain species and variants of *Bifidobacterium* are able to metabolize components of the early life diet, i.e., human milk oligosaccharides (HMO) present in breast milk, with *Bifidobacterium longum* subsp. *infantis* (9), *B. breve* (10), and *B. kashiwanohense* (11) enriched in the intestines of breast-fed infants. After weaning, *Bifidobacterium* ceases to predominate in the gut (12), and only species that can thrive on complex dietary carbohydrates are able to flourish. These species include *B. longum* subsp. *longum* (13, 14), *B. adolescentis,* and *B. pseudocatenulatum* (15).

B. pseudocatenulatum is less well characterized than other *Bifidobacterium* species but is associated with several health benefits. Expansion of *B. pseudocatenulatum* in the gut microbiome was associated with successful weight loss in obese children in China following ~100-day fiber-rich dietary (FRD) interventions (16, 17). According to a recent clinical trial, *B. pseudocatenulatum* was also among the enriched short-chain fatty-acid (SCFA)-producing gut commensals in type-2 diabetes patients receiving FRD interventions (18). Additionally, experimental evidence has demonstrated that supplementation with *B. pseudocatenulatum* CECT 7765 in obese mice led to improved metabolic responses (lowering serum cholesterol, triglyceride, and glucose concentrations) (19) and reduced proinflammatory cytokines (interleukin 17A [IL-17A] and tumor necrosis factor alpha [TNF-$\alpha$]) (20). Likewise, recent studies demonstrated that oral administration of *B. pseudocatenulatum* enhanced gut barrier integrity and alleviated bacterial translocation in mice with induced liver damage (21, 22). In a mouse colitis model, *B. pseudocatenulatum* oral supplementation also helped maintain the intestinal mechanical barrier and lowered proinflammation responses (23). These positive preclinical findings have promoted *B. pseudocatenulatum* development as a probiotic for clinical use. Recently, it has been proposed that a probiotic is defined as a bacterial strain satisfying the following criteria: (i) sufficiently characterized, (ii) safe for intended use, (iii) supported by at least one positive human clinical trial, and (iv) alive at an efficacious dose (24). So far, *B. pseudocatenulatum* CECT 7765 is the only strain of this species that fulfills these criteria. Its genome has been sequenced and characterized (25). A randomized controlled trial (RCT) in obese Spanish children with insulin resistance demonstrated that treatment with the probiotic *B. pseudocatenulatum* CECT 7765 at 10^9 to 10^10 CFU (colony forming unit) daily for 13 weeks (alongside dietary recommendation) resulted in a significant improvement in inflammatory status compared to that of dietary recommendation alone, which was reflected by a marked decrease in circulating C-reactive protein and increase in high-density lipoprotein cholesterol (26). Thus, the available data on this particular strain suggest that *B. pseudocatenulatum* holds potential probiotic traits which could ameliorate inflammation. However, future RCTs in human are needed to confirm this function, which would

need to be performed on a per-strain basis, with the additional requirement of further studies to meet strict regulatory standards (27).

In a recent microbiome study, we found that *B. pseudocatenulatum* was consistently depleted in the gut microbiomes of Vietnamese children suffering from dysenteric (mucoid and/or bloody) diarrhea compared to those suffering from watery diarrhea (28). This association remained significant regardless of etiological agent. Dysenteric diarrhea is associated with heightened inflammation, and we hypothesized that *B. pseudocatenulatum* may be beneficial in reducing inflammation-associated conditions and accelerating recovery of the gut microbiota following diarrhea. The health-promoting benefit of *B. pseudocatenulatum* is largely strain dependent and associated with distinct genome composition and biochemical profile (29). Locally sourced beneficial microbes represent an underexplored avenue for development of potential probiotics, which may be more compatible with the local food matrix and exhibit more ideal gut colonization and efficacy when administered to the target population (30). For instance, *Lactiplantibacillus plantarum* ATCC 202195, isolated from an Indian infant, was shown to colonize the neonatal gut for up to 4 months when orally administered with fructooligosaccharides (FOS) (31). This synbiotic (*L. plantarum* plus FOS) has proved successful in reducing the incidence of sepsis and death in rural Indian neonates, according to results published in a large-scale RCT (32). Therefore, aiming to generate data to support the development of a potential *B. pseudocatenulatum* probiotic suitable for use in treating/preventing dysenteric diarrhea, we assessed the genetic diversity of *B. pseudocatenulatum* colonizing the guts of healthy Vietnamese children and adults. Here, we defined the genetic diversity, predicted biochemical profile, and antimicrobial susceptibility of the *B. pseudocatenulatum* population in the Vietnamese population. These data are key for selecting an optimal probiotic candidate for downstream investigations and validation, with particular consideration for its potential uses in low-to-middle-income countries (LMICs) in Southeast Asia.

## RESULTS

**The prevalence of *Bifidobacterium pseudocatenulatum* in the Vietnamese population.** In order to investigate the distribution and diversity of *Bifidobacterium* spp. in the gut microbiota of Vietnamese adults and children, we extracted total DNA from fecal samples collected from 42 healthy Vietnamese individuals (21 children and 21 adults) and subjected them to shotgun metagenomic sequencing. All recruited children were aged between 9 and 59 months (median: 23 [interquartile range: 9 to 37] months) and had been weaned onto a solid food diet for at least 3 months. All recruited adults were aged between 25 and 59 years (median: 35 years) and reported to have an omnivorous diet. Taxonomic profiling from the microbiome data demonstrated that *Bifidobacterium* species were more abundant in children than in adults (median relative abundances: 8.0% [3.8 to 19.7] and 1.2% [0.3 to 3.4], respectively) (Fig. 1). Specifically, we found that *B. pseudocatenulatum* was the most prevalent *Bifidobacterium* species in the adults (mean = 1.1%) and the second-most prevalent *Bifidobacterium* species in children (mean = 2.9%, after *B. longum*).

**The phylogenetic distribution of Vietnamese *Bifidobacterium pseudocatenulatum*.** We selected fecal samples from participants with a relative *B. pseudocatenulatum* abundance of >0.1% (*n* = 16) to isolate *Bifidobacterium*. In total, we isolated 185 individual organisms with a colony morphology indicative of *Bifidobacterium*. Among these, 49 isolates (from 7 children and 6 adults) were *B. pseudocatenulatum* according to matrix-assisted laser desorption ionization–time of flight (MALDI-TOF) bacterial identification and full-length sequencing of the 16S rRNA (Fig. 1). These 49 individual organisms were subjected to whole-genome sequencing (WGS). A preliminary phylogenetic reconstruction using a core-genome alignment segregated the organisms into two distinct lineages (Fig. S1). The majority of isolates (*n* = 45) clustered with two *B. pseudocatenulatum* reference genomes (DSM20438 and CECT_7756) within the major lineage. The remaining four isolates (C01_H5, C01_D5, C01_C5, C02_A8) formed a separate cluster that was distantly

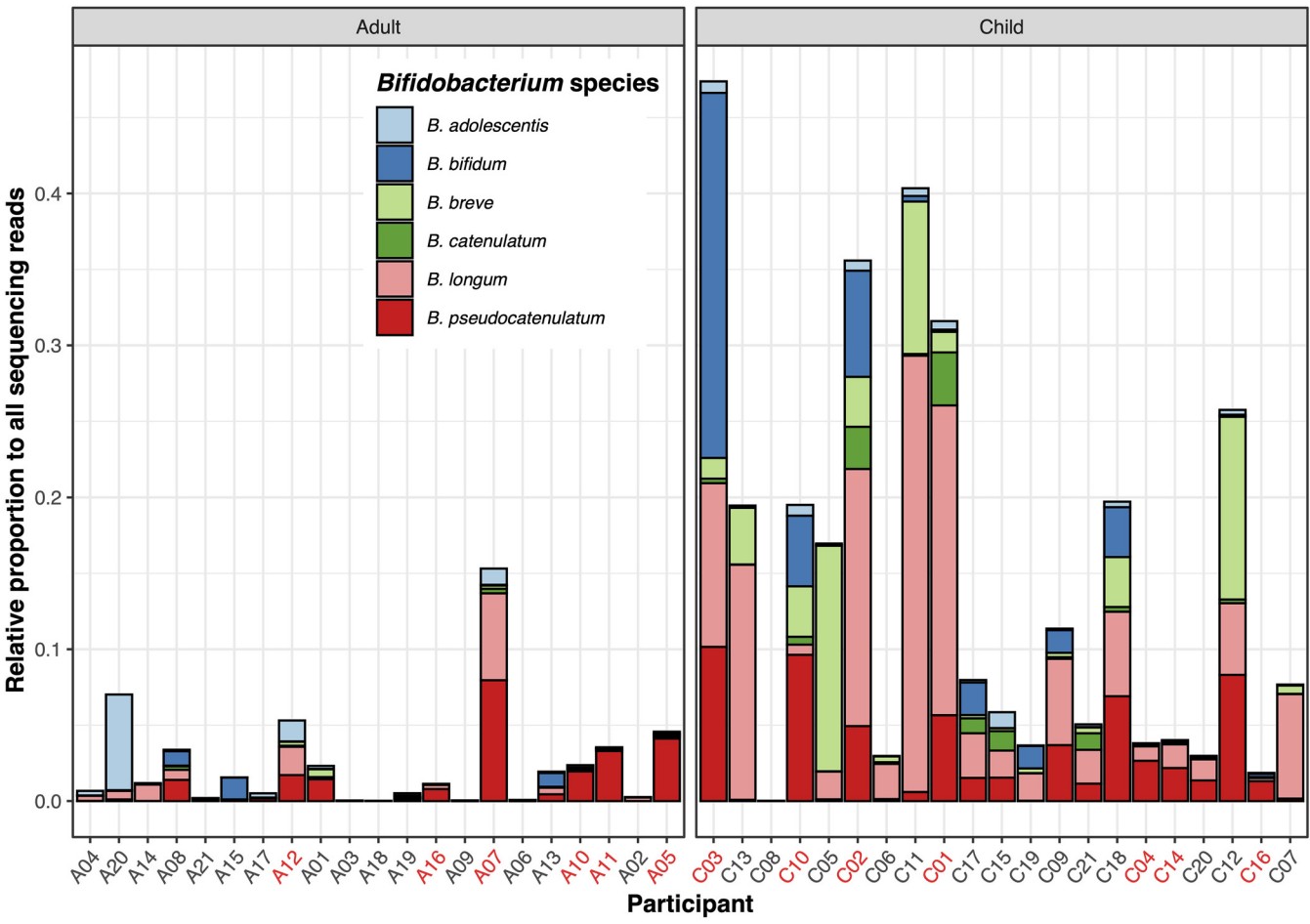

**FIG 1** Abundance of *Bifidobacterium* species in the gut microbiomes of a Vietnamese population. The figure displays the relative abundance of *Bifidobacterium* (calculated as percentage of reads classified as *Bifidobacterium* by Kraken, relative to each sample's total sequencing reads) in the gut microbiomes of adult (left) and child (right) participants (*n* = 21 for each group). The sample names in each group are ordered based on the participants' age, in increasing order. Samples labeled in red denote successful culture of laboratory-confirmed *B. pseudocatenulatum*. Different *Bifidobacterium* species are colored as in the key.

related to the major lineage. Further interrogation and comparison with *B. catenulatum* and *B. kashiwanohense* genomes confirmed that these four isolates belonged to the *B. catenulatum* spp. complex (Fig. S2).

Refined phylogenetic reconstruction of the 45 *B. pseudocatenulatum* genomes identified 13 phylogenetic clusters (PCs) and three singletons (C14_S, A05_S, A16_S) (Fig. 2), all of which were supported by high bootstrap values (>80%). For ease of nomenclature, these PCs and singletons were collectively called PCs. Each PC was defined by close genetic relatedness (negligible branch lengths), and each contained organisms isolated exclusively from a single individual. However, isolates recovered from each sampled participant were either solely restricted to one PC (6/11 participants) or distributed across two PCs (5/11). Moreover, when multiple PCs were detected within the same individual, they were generally not monophyletic (except for C16). These data suggest that the *B. pseudocatenulatum* population within each individual is highly diverse, and PCs could not be distinguished based on the age of the participant (child versus adult). Additionally, phylogenetic reconstruction of global *B. pseudocatenulatum* isolates indicated the presence of geographical clustering, such as that observed in *B. pseudocatenulatum* isolated from China, Japan, and Vietnam (Fig. 3). However, the lack of high bootstrap support at the basal nodes rendered this observation inconclusive.

**In silico prediction of carbohydrate utilization.** *Bifidobacterium* spp. are renowned for their ability to utilize a diverse range of carbohydrates, which contribute to the functional integrity of the human gut. We focused on identifying the repertoire of

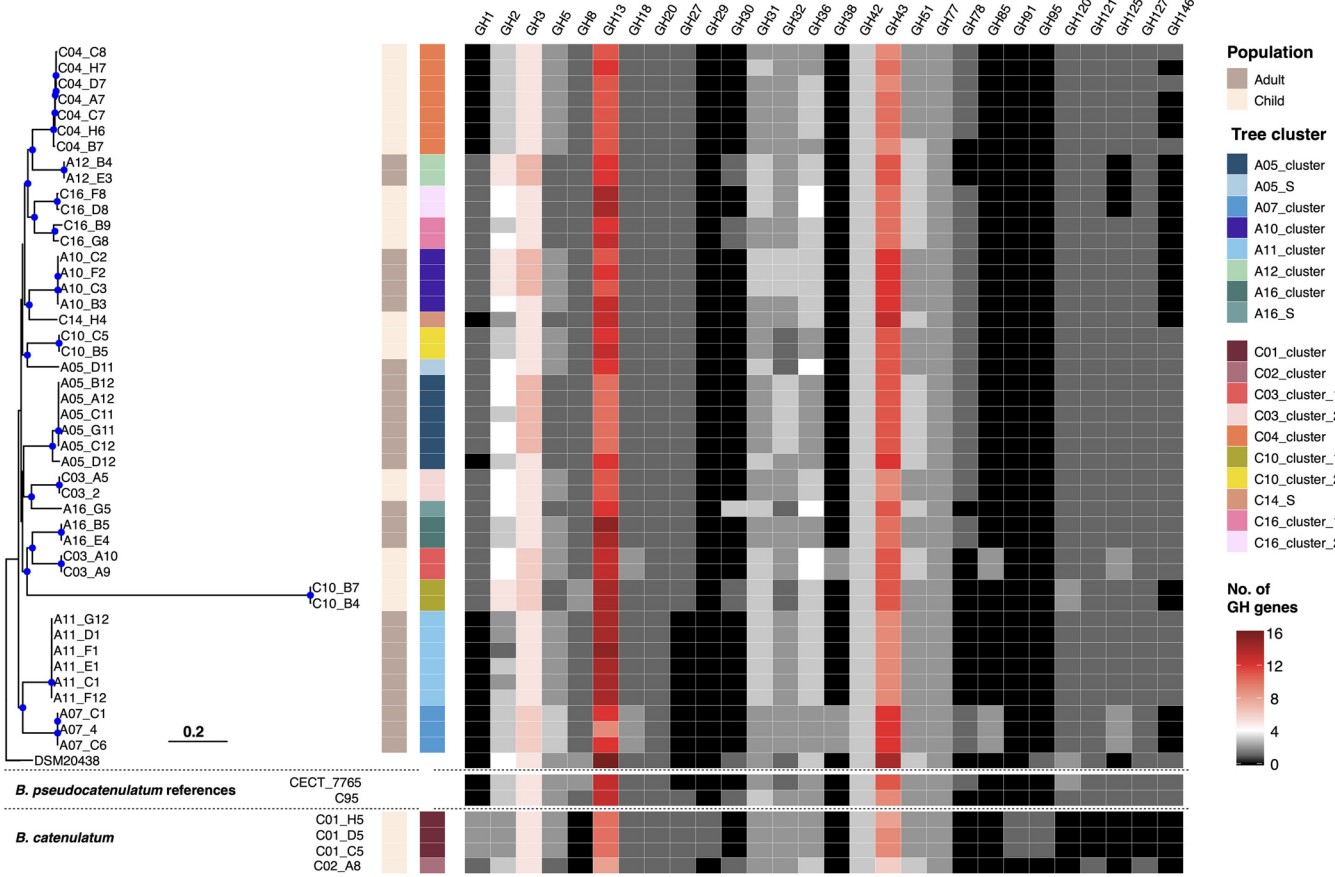

**FIG 2** The phylogeny and glycosyl hydrolase profile of Vietnamese *Bifidobacterium pseudocatenulatum*. The maximum likelihood phylogeny of 45 *B. pseudocatenulatum* isolates in this study was constructed from a recombination-free alignment following read mapping (see Materials and Methods). The tree is rooted using the reference DSM20438 as an outgroup, and blue filled circles denote bootstrap values greater than 70 at the internal nodes. Other *B. pseudocatenulatum* references (CECT_7765 and the Chinese isolate C95) and *B. catenulatum* isolated in this study are included for comparison. The columns to the right of the phylogeny show the metadata associated with each taxon, including population, tree cluster nomenclature, and the numbers of the genes belonging to each defined glycosyl hydrolase family (GH1, GH2, GH3, GH5, GH8, GH13, GH18, GH20, GH27, GH29, GH30, GH31, GH32, GH36, GH38, GH42, GH43, GH51, GH77, GH78, GH85, GH91, GH95, GH120, GH121, GH125, GH127, GH146). The quantity of these genes is denoted according to the key. The horizontal scale bar denotes the number of substitutions per site.

carbohydrate-utilizing enzymes (CAZymes) within the *B. pseudocatenulatum* genome sequences to predict the carbohydrate metabolic capacity of each isolate. Among 4,333 gene families in the pangenome, 233 were determined to be CAZymes. These included 126 glycosyl hydrolases (GH), 97 glycosyltransferases (GT), 2 carbohydrate esterases, and 2 carbohydrate-binding motif (CBM)-containing proteins. As GHs catalyze the breakdown of glycosidic bonds, they are essential for the assimilation of complex glycans. We mapped the presence of all GH genes in each isolate of the *B. pseudocatenulatum* (45 Vietnamese and 7 reference isolates) and *B. catenulatum* (C01 and C02 clusters) collections. Genes pertaining to GH23 and GH25 were excluded from interpretation, as they participate specifically in the recycling of the peptidoglycan in the bacterial cell wall.

Thirty-four GH genes were classified as core (present in all 52 *B. pseudocatenulatum* genomes), while accessory GH genes were more enriched in ≤10 genomes (Fig. S3). The predominant GH families identified were GH13 (median of 12 copies per isolate) and GH43 (median of 10.5 copies per isolate), followed by GH3 (median of 5 copies per isolate) (Fig. 2). GH13 catalyze mainly the hydrolysis of $\alpha$-glucosidic linkages (in resistant starch and $\alpha$-glycans), while GH3 is involved in the assimilation of cellobiose and cellodextrin. GH43 includes a diverse range of $\alpha$-L-arabinofuranosidase, $\beta$-xylosidase, and xylanase involved in the degradation of hemicellulose, arabinogalactan, arabinan, and arabinoxylan. Other common GHs expand *B. pseudocatenulatum*'s catabolism range

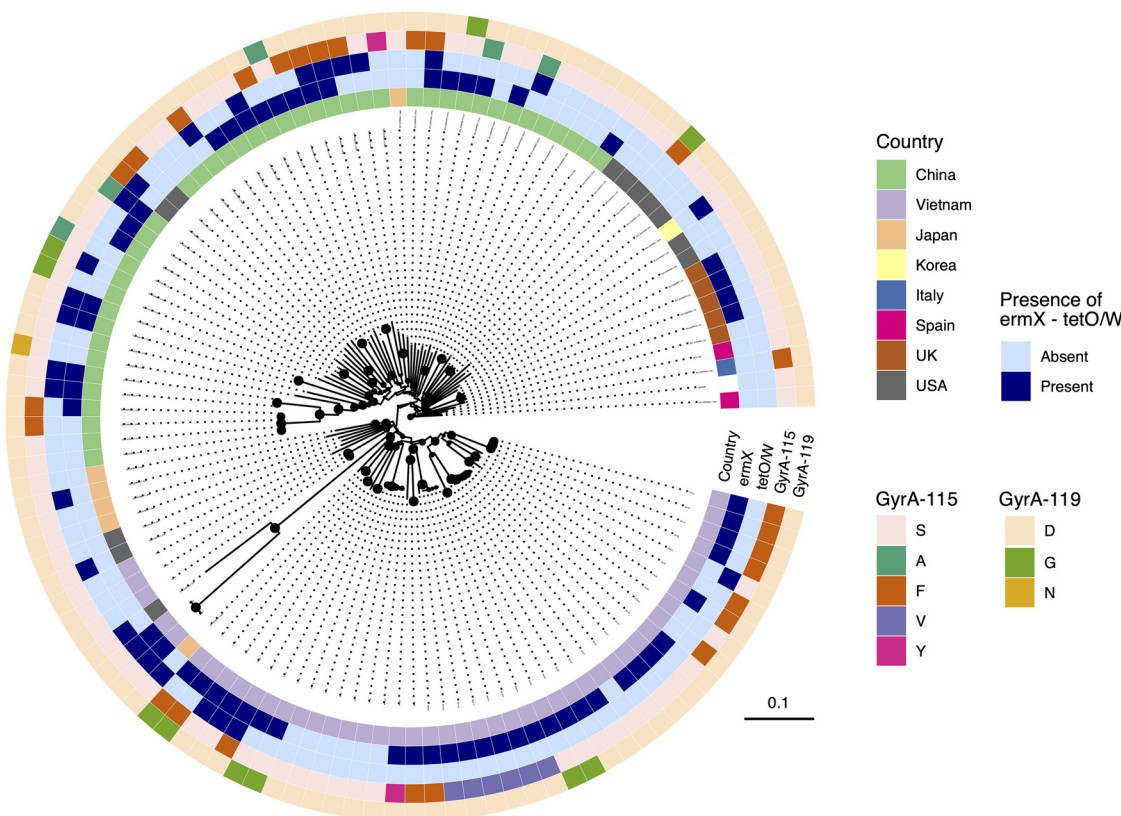

**FIG 3** Phylogenetic reconstruction of global *Bifidobacterium pseudocatenulatum*. The maximum likelihood phylogeny of 116 *B. pseudocatenulatum* (isolated in this study and accessed via NCBI) was constructed from a recombination-free alignment following core-genome reconstruction (see Materials and Methods). The tree is rooted using the reference DSM20438 as an outgroup, and the size of black filled circles is proportional to bootstrap values at the internal nodes. Associated metadata were appended on the phylogeny, including the country of isolation, presence of *ermX* and *tetO/W*, and quinolone resistance-determining region (QRDR) specific mutations (GyrA-115 and GyrA-119) (see legends). The horizontal scale bar denotes the number of substitutions per site.

toward beta-galactosides/lactose (GH1, GH2, and GH42), fructan (GH32), xylan (GH5, GH8, GH31, GH120, and GH30), raffinose/melibiose (GH27 and GH36), chitin (GH18), amylose (GH77), rhamnosides (GH78), and arabinoxylan/arabinogalactan (GH51, GH121, GH127, and GH146). In contrast, GH targeting HMOs or mucin (GH29, GH33, GH35, and GH95) were not detected in this *B. pseudocatenulatum* collection, except for the N-linked glycans specific mannosidases (GH38 and GH125) and GH85. These data signify a tropism for dietary starch and fiber in the catabolism of these organisms.

**Genomic variation within the phylogenetic cluster.** Isolates within each PC could share limited or substantial variation in the core and accessory genomes. Within each PC, inter-isolate variation in the core genome (pairwise recombination-free single nucleotide polymorphisms [SNPs]) closely mirrors that inferred from the pangenome (presence/absence of accessory genes) (Fig. S4 and Fig. 4A). This led us to hypothesize that the extent of intra-PC variation in the pangenome was dependent on the evolutionary time frame of the PC, which is reflected in the number of core-genome SNPs the PC has accumulated since its most recent common ancestor (MRCA). Within the examined PCs, the median of pairwise recombination-free SNPs was 9 (interquartile range [IQR] 13 to 49), while the median variation in the pangenome was 24 genes (IQR 13 to 97). As shown in Fig. 4B, the pairwise difference in the accessory genome correlated partially with the pairwise SNP distance (Pearson's $r = 0.54$). Outliers to this trend were detected in the C16_cluster_1 (from a 57-month-old child) and A05_cluster (from a 59-year-old female), in which ~400 SNP distance was noted, albeit sharing only ~100 gene differences in the accessory genome.

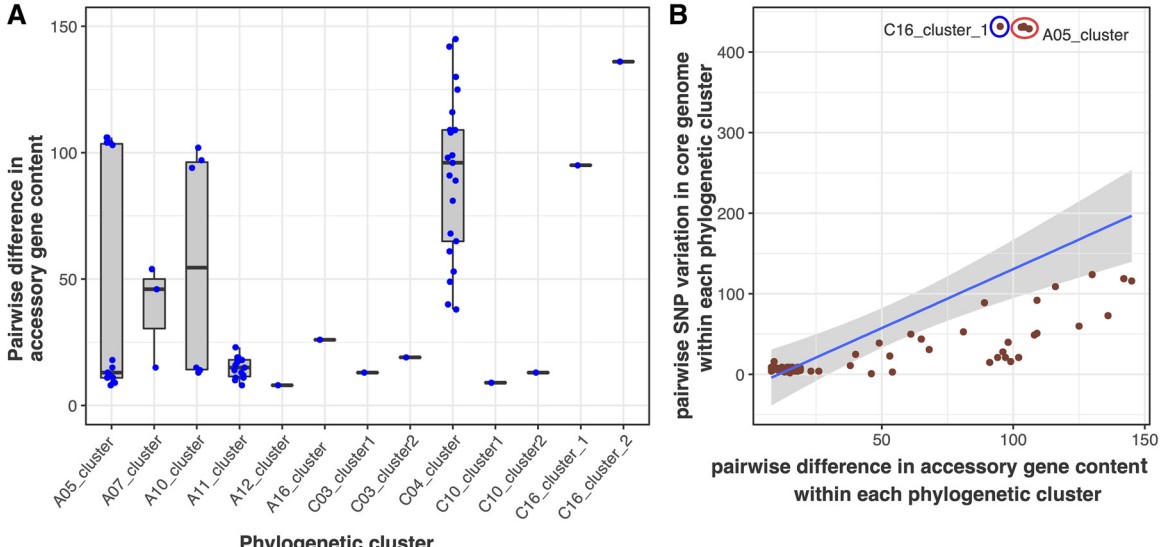

**FIG 4** Variation in the accessory genomes of *Bifidobacterium pseudocatenulatum*. (A) The panel depicts the distribution of pairwise differences in the accessory gene content (counted as presence/absence, represented as blue circles) of isolates within each defined tree cluster. For each box plot, the upper whisker extends from the 75th percentile to the highest value within 1.5× the interquartile range (IQR) of the hinge, and the lower whisker extends from the 25th percentile to the lowest value within 1.5× the IQR of the hinge. (B) The panel displays the positive correlation between intraclonal pairwise differences in the accessory gene content (*x* axis) and intraclonal pairwise variation of single nucleotide polymorphisms (SNPs) in the core genome. The blue and red circles indicate the outliers, C16_cluster_1 and A05_cluster_1.

Detailed characterization revealed that inter-isolate variation was found to be minimal for 7/13 nonsingleton PCs (≤16 SNPs in core and ≤26 differences in accessory) (Fig. 3A), indicating that the isolates were likely clonal and belonged to the same strain. This limited genetic diversity may arise due to errors in analytical processes (read mapping, assembly, and annotation) or may be attributed to insufficient sampling coverage; however, more intensive sampling, as demonstrated in the A11_cluster (six isolates), still resulted in a low variation. The differences in intraclonal gene content were typically associated with genes encoding carbohydrate transport (ABC transporter and permease) and utilization proteins (GH, GT, esterase) or were of unknown function (6 to 25 hypothetical proteins per PC). Alternatively, the bimodal distribution observed in the A05_cluster, A07_cluster, and A10_cluster demonstrates that while most organisms share limited variation in gene content (~12 genes), outlier organisms may carry a distinct accessory genome. This observation resulted in sizeable differences when comparing the outlier to the remainders in each PC. For example, the accessory genome of A05_D12 (A05_cluster) differed from that of the remaining five isolates by >100 genes. This composition of genes arose from a recombination event (spanning 28 kbp from 2,126,733 to 2,154,962 in the DSM20438 chromosome and containing multiple ABC transporters and GHs) and the gain of an IS3-mediated region (ABC transporters and β-glucosidase), which distinguished A05_D12 from the other isolates (Fig. S5). Likewise, recombination in the same region and the lipopolysaccharide (LPS) biosynthesis cluster resulted in a significant (~100 genes) difference between A10_B3 and the remaining A10_cluster isolates (Fig. S6). In A07_cluster, A07_4 lost a genomic region (spanning 48 kbp from 78,148 to 126,915 on DSM20438, containing genes encoding branched-chain amino acid and carbohydrate transport systems, fimbrial subunit FimA, GHs, and thiamine biosynthesis ThiSGF) compared to the other two isolates.

Most noticeably, the C04_cluster contained a wider distribution of pairwise differences across the core and accessory genome, indicating that each isolate was a distinct strain and possessed a moderately divergent accessory genome. To visualize the

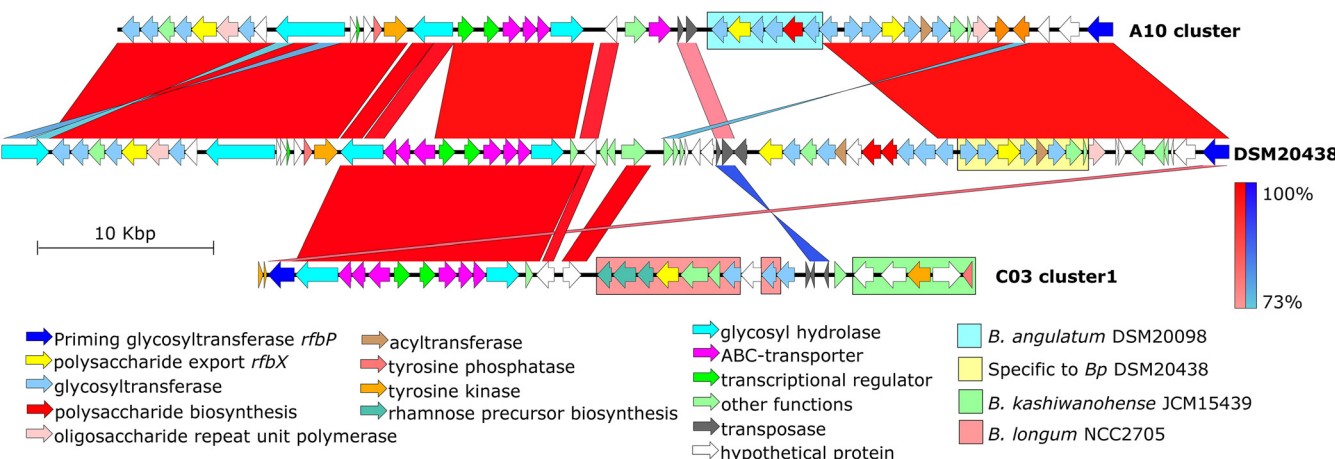

**FIG 5** Comparison of the exopolysaccharide (EPS) biosynthesis genomic region from exemplar *Bifidobacterium pseudocatenulatum* genomes. Each arrow represents a predicted gene, with its function colored as in the keys. The blocks connecting two genomes indicate regions with high nucleotide similarity, as either synteny (red) or inversion (blue), with the color intensity corresponding to the degree of nucleotide similarity. The colored boxes denote regions homologous to that found in specific *Bifidobacterium* species.

magnitude of lateral gene transfer, we separately reconstructed the phylogeny of the C04_cluster (Fig. S7). Branch lengths indicative of significant divergence, coupled with a high frequency of gene acquisition and gene loss events, demonstrate that the C04 *B. pseudocatenulatum* population has undergone extensive clonal expansion. This microevolution underlies a diversifying metabolic potential, as exemplified by the concurrent acquisition of a GH78 (rhamnosidase) and deletion of a GH51 (arabinosidase) in a specific monophyletic cluster (Fig. 2 and Fig. S7). Notably, a novel ribulose-5P-3-epimerase gene was acquired in the most recent common ancestor (MRCA) of C04_A7 and C04_D7. This enzyme bilaterally converts ribulose-5P to xylulose-5P, a key intermediate in the *Bifidobacterium* specific hexose metabolic pathway (bifid shunt) (8), thus potentially facilitating a greater energy harvest. We aimed to identify the genetic origin of the acquired elements (by BLAST to public database) and found that *B. kashiwanohense* and *B. adolescentis* were the most likely sources. Overall, the presented evidences suggest that as the *B. pseudocatenulatum* population undergoes a prolonged period of within-host evolution and expansion, its pangenome may expand mainly through increasing horizontal gene transfer (HGT) with other *Bifidobacterium* species. The usual targets of such events involve genes responsible for carbohydrate transport and metabolism, which expedite the diversification of clonal bacteria.

**Genomic differences in *Bifidobacterium pseudocatenulatum* originating from children and adults.** The differing physiologies and diets of children and adults create distinct niches in which *Bifidobacterium* can adapt, and such adaptation may be reflected by genomic variation. An exploratory analysis of 21 representative isolates (10 from adults, 11 from children) identified 42 genes with differing abundance between *B. pseudocatenulatum* isolates originating from children and those originating from adults. Of these 42 genes, 13 were of unknown function. Eight genes (4 glycosyltransferases, a polysaccharide export *rfbX*, an O-acetyltransferase, a reductase, and *fhiA*) were present in bacteria retrieved from adults more frequently (6/10 representative clusters) than in those from children (1/11 representative clusters) (Fisher's exact test, $P = 0.024$). These genes formed the core component of the exopolysaccharide (EPS) biosynthesis cluster of the reference strain DSM20438 (Fig. 5). We characterized this genomic region of all representative *B. pseudocatenulatum* in detail and confirmed that the EPS region in organisms from adults was more similar to that of DSM20438 (Table 1). In organisms isolated from both adults and children, the EPS cluster was subject to frequent modification, with integrations of genes derived from the EPS region of other *Bifidobacterium* species, such as *B. longum* and *B. kashiwanohense*. Notably, the rhamnose precursor biosynthesis genes (*rmlABC*) were present in isolates of three clusters (C03_cluster_1, A07_cluster, A16_cluster), which

**TABLE 1** Exopolysaccharide biosynthesis of *B. pseudocatenulatum*[a]

| Group | Phylogenetic cluster | Presence of *B. pseudocatenulatum* DSM20438 specific EPS genes | Variable region | Rhamnose biosynthesis genes present |
|---|---|---|---|---|
| Ref | CECT_7765[b] | | *B. longum* NCC2705 (*rfbX*, GTs) and *B. kashiwanohense* JCM15439 | *rmlABC* |
| Child | C03_cluster_1[b] | | *B. longum* NCC2705 (*rbfX*, GTs) and *B. kashiwanohense* JCM15439 | *rmlABC* |
| | C03_cluster_2 | | *B. breve* lw01 (*rbfX*, GTs, OAL) | |
| | C04_cluster | | *B. gallinarum* CACC514 (*rfbX*, GTs) and *B. kashiwanohense* PV20-2 | |
| | C10_cluster_1 | | Uncharacterized (*rfbP*, *cpsD*, *rfbX*, GTs) | |
| | C10_cluster_2 | | Uncharacterized (*rfbX*, GTs, AT) | |
| | C14_S | Yes | Uncharacterized (*rfbX*, GTs, AT) | |
| | C16_cluster_1[c] | | *B. breve* JCM7017 (*rfbX*, GTs) and uncharacterized (*rfbX*, GTs, AT) | |
| | C16_cluster_2[c] | | *B. breve* JCM7017 (*rfbX*, GTs) and uncharacterized (*rfbX*, GTs, AT) | |
| Adult | A05_cluster | | *B. kashiwanohense* PV20-2 (*rbfX*, GTs) and *B. longum* NCTC11818 (GTs, OAL) | |
| | A05_S[d] | Yes | Uncharacterized (*rfbX*, GTs, AT) | |
| | A07_cluster | | *B. longum* 105-A (*rfbX*, OAL, GTs) | *rmlABC* |
| | A10_cluster[e] | Yes | *B. angulatum* DSM20098 (*rfbX*, GTs, OAL) | |
| | A11_cluster[d] | Yes | Uncharacterized (*rfbX*, GTs, AT) | |
| | A12_cluster | Yes | Uncharacterized (*rfbX*, GTs, AT) | |
| | A16_cluster | | *B. longum* ZJ1 (*rfbX*, GTs, OAL) | *rmlABC* |
| | A16_S[e] | Yes | *B. angulatum* DSM20098 (*rfbX*, GTs, OAL) | |

[a]GT, glycosyltransferase; AT, acyl-transferase; OAL, O-antigen ligase; *rfbX*, O-antigen transporter. *rfbP*, *cpsD*, priming glycosyltransferase. Isolates with the same superscript letter [b, c, d, and e] share similar EPS biosynthesis cluster.

predicts the incorporation of rhamnose or rhamnose-derived sugar in the EPS structure (33). Organisms of distantly related PCs occasionally shared comparable EPS regions, as observed in A05_S and the A11_cluster. Specifically, the EPS region of C03_cluster_1 was similar to that of *B. pseudocatenulatum* CECT_7765, which has been developed as a probiotic candidate to alleviate inflammatory responses in patients with cirrhosis and obesity in Spain (26, 34).

Two additional genes were found to be enriched in *B. pseudocatenulatum* isolated from adults (8/10) compared to those from children (3/11) (Fisher's exact test, $P = 0.03$). These two tandem genes (BBPC_RS09395 and BBPC_RS08115 in the reference DSM20438) both encode GH43. RS09395 is a large multidomain protein (>2,000 amino acids [aa]) and consists of three GH43 subunits. Among these, two subunits shared >70% nucleotide identity with the $\alpha$-L-arabinofuranosidases, *arafB* (BLLJ_1853, GH43_22) and *arafE* (BLLJ_1850, GH43_34), of *B. longum* JCM1217, which encode degradative enzymes targeting the arabinan backbone and arabinoxylan, respectively (35). The remaining GH43_22 subunit of RS09395 showed no ortholog in *B. longum* and shared 65% amino acid identity with that in *B. catulorum*. In contrast, RS08115 was smaller (~1,000 aa) and shared 65% nucleotide identity with *arafA* (BLLJ_1854, GH43_22) of *B. longum* JCM1217, known to specifically degrade arabinogalactan (36). Bioinformatic analyses predicted that both RS09395 and RS08115 were secreted and bound to the bacterial cell membrane, due to the presence of N-terminal signal peptide and transmembrane motifs. These data suggest that these two enzymes synergistically degrade arabinoglycan, releasing degradants (i.e., L-arabinose) into the extracellular milieu and contributing to cross-feeding with other members of the gut microbiota.

**Antimicrobial susceptibility of representative *Bifidobacterium pseudocatenulatum*.** To better evaluate their suitability for potential probiotic design, specifically to assess if they can be formulated along with antimicrobial treatments, we subjected the isolated *B. pseudocatenulatum* to antimicrobial susceptibility profiling. We reported a broad range of MICs against ceftriaxone, amoxicillin/clavulanate, ciprofloxacin, azithromycin,

**TABLE 2** Summary of Etest results for 21 *Bifidobacterium* isolates (19 representative *B. pseudocatenulatum* isolates and 2 controls)[a]

| Antimicrobial | No. of strains with MIC (µg/ml): | | | | | | | | | | | | | | | | | | | | | | |
|---|---|---|---|---|---|---|---|---|---|---|---|---|---|---|---|---|---|---|---|---|---|---|---|
| | ≤0.047 | 0.064 | 0.094 | 0.125 | 0.19 | 0.25 | 0.38 | 0.5 | 0.75 | 1 | 1.5 | 2 | 3 | 4 | 8 | 12 | 16 | 24 | 32 | 48 | 64 | 128 | >256 |
| CRO | | | 2 | 4 | 2 | 4 | 1 | 3 | 2 | | 1 | | | | | | | | | | | | | |
| CIP | | | | | | | | 1 | 1 | 3 | 1 | 2 | | | | | | | 11 | | | | |
| AMC | 2 | 2 | 4 | 7 | 2 | 2 | | | | | | | | | | | | | | | | | |
| AZM | 1 | | 1 | | 1 | 1 | 1 | 2 | 2 | 1 | | 1 | | 1 | | | | | | 1 | | 1 | 5 |
| TET | | | | | | | 2 | 1 | 4 | 4 | | 2 | 1 | 1 | | | 1 | | | 2 | 1 | | |
| MTZ | | | | | | | | | | | | 1 | | | | 1 | 1 | 1 | | 2 | 1 | 1 | 11 |

[a]CRO, ceftriaxone (30 µg); CIP, ciprofloxacin (5 µg); AMC, amoxicillin-clavulanic acid (30 µg); AZM, azithromycin (15 µg); TET, tetracycline (30 µg); MTZ, metronidazole (5 µg).

tetracycline, and metronidazole for 19 representative isolates (17 *B. pseudocatenulatum* isolates, 2 *B. catenulatum* isolates) (Table 2). Notably, the MICs for ceftriaxone (≤1.5 µg/ml) and amoxicillin/clavulanate (≤0.25 µg/ml) were consistently low, likely indicating that all tested *Bifidobacterium* would be susceptible to these β-lactams during antimicrobial therapy. In contrast, susceptibility against the remaining antimicrobials was more variable, as evidenced by their broader MIC ranges. The highest MICs for ciprofloxacin (32 µg/ml) and metronidazole (256 µg/ml) were observed in 57% (11/19) of tested *Bifidobacterium* isolates (Table 2). Concurrent nonsusceptibility against ciprofloxacin (MIC = 32 µg/ml), azithromycin (MIC = 256 µg/ml), and metronidazole (MIC = 256 µg/ml) was observed in four isolates.

We examined the correlation between MIC values and inhibitory zone diameters (IZD) for the six aforementioned antimicrobials (Fig. S8). The narrow range of recorded values for amoxicillin/clavulanate (0.047 to 0.25 µg/ml, 36 to 48 mm) and ceftriaxone (0.094 to 1.5 µg/ml, 28 to 40 mm) resulted in a weak to modest negative correlation (Kendall's $r \geq -0.5$). Such correlation appeared to be stronger for azithromycin, tetracycline, and ciprofloxacin (Kendall's $r \leq -0.7$), likely owing to a wider range of MIC and IZD values. Specifically, for ciprofloxacin, an MIC value of 32 µg/ml corresponded with an IZD of 6 mm (no killing zone), while the remaining MIC (0.5 to 2 µg/ml) corresponded with IZDs of >18 mm. In contrast, a poorer correlation was observed with metronidazole despite presenting a similarly broad range of MIC values, such that an IZD of 6 mm corresponded with a wide range of MICs (12 to 256 µg/ml). These results showed that, with exception of metronidazole, both Etest and disk diffusion methods produce robust and consistent interpretations for antimicrobial susceptibility in these *Bifidobacterium* species.

**Antimicrobial resistance genetic determinants in *Bifidobacterium pseudocatenulatum*.** We last sought to investigate potential mechanisms of antimicrobial resistance (AMR) in the sequenced *Bifidobacterium* genomes. As resistance to metronidazole is complex and typically attributed to altered metabolism (37), we focused only on the genetic determinants for resistance against tetracycline, azithromycin, and ciprofloxacin. Screening against a curated database of acquired AMR genes revealed the presence of *tetO* and *ermX* in our isolates. The presence of *tetO* correlated significantly with elevated MIC and decreased IZD against tetracycline, while the presence of *ermX* was associated with a decrease in azithromycin IZD (Fig. 6). As ciprofloxacin resistance is commonly induced by mutations in the quinolone resistance-determining region (QRDR) on bacterial topoisomerases (38–40), we screened *gyrA*, *gyrB*, *parC*, and *parE* in the assembled *Bifidobacterium* genomes to identify nonsynonymous mutations in the QRDR. This analysis detected the presence of nonsynonymous mutations in *gyrA*. These included single mutations such as S115F ($n = 5$), S115V ($n = 1$), S115Y ($n = 1$), and D119G ($n = 3$), as well as a double mutation, S115F D119G ($n = 1$). Upon classifying the isolates based on the presence of the aforementioned mutations, we found that their presence correlated significantly with elevated MIC (32 µg/ml) and reduced IZD (6 mm) (Fig. 6). These data indicate that the presence of *tetO*, *ermX*, and specific mutations in *gyrA* in our *Bifidobacterium* collection explain nonsusceptibility against tetracycline, azithromycin, and ciprofloxacin, respectively. Furthermore, we expanded screening

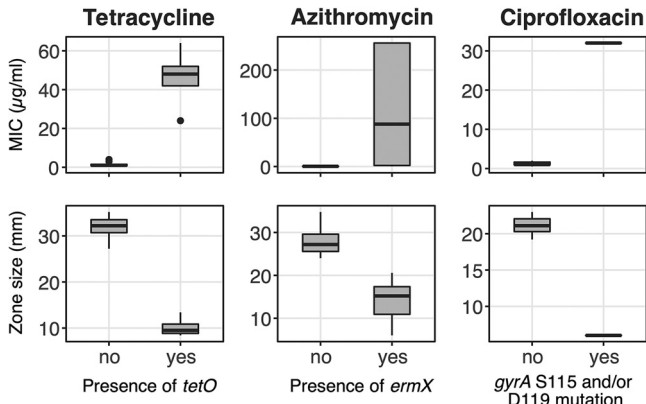

**FIG 6** Association between resistance determinants and antimicrobial testing results in *Bifidobacterium*. Each column displays the Etest (MIC in μg/ml) and disk diffusion (inhibitory zone diameter in mm) results for a tested antimicrobial, classified based on the presence of target resistance determinants (*tetO*, *ermX*, and *gyrA* S115 and/or D119 mutation).

for these AMR determinants in the compiled global *B. pseudocatenulatum* isolates collection. Particularly, for 33 nonclonal *B. pseudocatenulatum* isolates from China, the prevalence of *ermX*, *tetO/W*, and QRDR mutations was found to be 54.5% (18/33), 33.3% (11/33), and 54.5% (18/33), respectively, similar to our findings in Vietnam (Fig. 3). In contrast, *B. pseudocatenulatum* cultures isolated from developed countries (Italy, Japan, South Korea, Spain, United Kingdom, United States) showed a lower prevalence in these elements (*ermX*: 1/22; *tetO/W*: 6/22; QRDR single/double mutations: 4/22). These indicate that resistances to azithromycin and fluoroquinolone could be more commonplace in *B. pseudocatenulatum* isolated from Asian developing nations, likely owing to recent heightened usage of and easy access to these antimicrobials (41).

## DISCUSSION

Our study is among the first to use the combination of anaerobic microbiology and genomics to study the diversity and antimicrobial susceptibility of a *Bifidobacterium* species in a developing country setting. We found that the *B. pseudocatenulatum* population within each individual is distinct and diverse. Intraclonal variation in the pangenome usually stems from the gains or losses of glycosyl hydrolases and associated carbohydrate transporters, thus creating divergent metabolic functions, even for isolates within the same evolving clone. Our results reiterate previous findings on the genomic characterization of the *Bifidobacterium* genus (7) and *B. pseudocatenulatum* in the European population (42), showing that the species harbors an expansive repertoire of enzymes (GH13, GH43, and other GHs) responsible for the assimilation of complex plant-derived carbohydrates but not host-derived glycan (mucin, HMO). In line with this observation, experimental study showed that *B. pseudocatenulatum* could utilize several components of arabinoxylan hydrolysates (AXH) via the upregulation of three GH43-ABC transporter clusters (43). This feature likely explains the abundance and persistence of *B. pseudocatenulatum* through adulthood in the Vietnamese gut microbiota, since the organism may thrive on nondigestible carbohydrates enriched in fruits and vegetables. Additionally, we also found that *B. pseudocatenulatum* is likely phylogenetically clustered based on geography, with distinct *B. pseudocatenulatum* population associated with each country or ethnicity. This observation is similar to the phylogenetic structure of other human bacteria with vertical mode of transmission, such as *Helicobacter pylori* (44). Such local expansion could render *Bifidobacterium* sufficient time to evolve and adapt to local diets, which is subject to future investigations in more large-scale studies.

We found that enzymes homologous to *B. longum* α-L-arabinofuranosidases (ArafA, ArafB, and ArafE) are potentially more abundant in adult-derived *B. pseudocatenulatum*,

which reflects the adaptation of the species to the more complex fiber-rich diet usually present in adulthood. Similarly, a microevolution study of *B. longum* subsp. *longum* in Japan highlighted the enrichment of these homologs in strains derived from an elderly population (13). The above evidence could suggest a pattern of convergent evolution across different *Bifidobacterium* species, indicative of adaptation to resource availability. As these enzymes were predicted to be secreted in our *B. pseudocatenulatum* isolates, they may facilitate cross-feeding pathways with other bacteria incapable of utilizing complex arabinoglycan hydrolysates. Metabolic cross-feeding has been noted in *Bifidobacterium*, in which the fermentation end products lactate and acetate can be utilized by anaerobes *Eubacterium hallii* (45) and *Faecalibacterium prausnitzii* (46), respectively, to produce butyrate. Likewise, *B. pseudocatenulatum* strains capable of degrading HMO were shown to release simpler metabolites, which supported the growth of other *Bifidobacterium* strains from the same breast-fed individual (42).

The EPS biosynthesis region was present in all recovered *B. pseudocatenulatum* genomes; this region is subjected to a high degree of genetic modification. The genetic composition of the EPS region is potentially more diverse in child-derived *B. pseudocatenulatum*, possibly owing to a greater extent of HGT in the *Bifidobacterium*-predominant gut microbiota in children. Previous studies have found that the EPS structure is critical for elucidating the *Bifidobacterium*-host interaction. In a simulated human intestinal environment, genes related to EPS biosynthesis were substantially upregulated (47). The surface EPS grants protection against low pH and bile salt in the gastrointestinal environment (48, 49). Moreover, the presence of EPS in *B. breve* was associated with lower proinflammatory cytokines and antibody responses, which facilitate its persistent colonization in mouse models (48). However, it is known that *Bifidobacterium* strains with different EPS structures, even within a single species, can elicit differing immunological responses *in vitro* (50). For example, high-molecular-weight EPS is more likely to induce weaker immune responses, potentially because these encapsulating structures shield the complex protein antigens on the bacterial surface from interaction with immune cells (51). Furthermore, a specific EPS from *B. breve* has been shown to be metabolized by some members of the infant gut microbiota, indicating that EPS further facilitates cross-feedings between gut bacteria (52). Therefore, the considerable diversity shown by different EPS genomic characterizations in our *B. pseudocatenulatum* collection suggests the induction of varying and strain-dependent immunological responses within the human host. Recently, rhamnose-rich EPS were shown to elicit a moderate secretion of proinflammatory cytokines, prompting a mild boost to innate immunity (53). The EPS genomic clusters of the well-researched probiotic *B. pseudocatenulatum* CECT 7765, which has demonstrated a variety of health-promoting traits (26, 34), and several of our *B. pseudocatenulatum* isolates carried the *rmlABC* locus responsible for rhamnose biosynthesis. Future studies should investigate whether different rhamnose-rich EPS in *Bifidobacterium* confer similar effects on inflammatory responses and ultimately on host health.

Our findings concur with previous studies on AMR in *Bifidobacterium*, showing that the carriage of *ermX* (54) and *tet* variants (55, 56) is common in the genus. These elements induce decreased susceptibility to azithromycin and tetracycline, respectively, which we confirmed in our study and has been observed in previous investigations (57). We also report a high prevalence of metronidazole resistance in these Vietnamese *B. pseudocatenulatum* isolates. Metronidazole resistance in *Bifidobacterium* has been observed occasionally (58) and was recently reported in *B. pseudocatenulatum* strains causing pyogenic infections (59), but the resistance mechanism remains elusive. In addition, we showed here that mutations in the QRDR of *gyrA* (S115 and/or D119) are likely to stimulate increased resistance to ciprofloxacin in *Bifidobacterium*. The encompassing region SAIYD (position 115 to 119 in GyrA) in wild-type *B. pseudocatenulatum* corresponds to the conserved SA[**X**]YD (83 to 87) in *Escherichia coli*'s GyrA, the most well-studied QRDR (40). However, unlike *E. coli*, which requires triple mutations (two in *gyrA*, one in *parC*) for full ciprofloxacin resistance ($\geq 2$ $\mu$g/ml), a single mutation in *B. pseudocatenulatum* may elevate ciprofloxacin MIC to $\geq 32$ $\mu$g/ml ($>$16-fold increase). The

ease of such a single-step process may explain the high degree of independent resistant mutations across different clones and geographies. Ciprofloxacin, azithromycin, and metronidazole are frequently prescribed for treatment of various infections in Vietnam. Prolonged exposure to these antimicrobials could induce resistance in *Bifidobacterium*, facilitated by the mobility of resistance determinants in the gut microbiota environment (56, 60). Though several studies have investigated the disturbance and recovery of the gut microbiome postantibiotic treatment (61, 62), it is unknown how AMR in specific gut commensals (i.e., *Bifidobacterium*) affects these ecological trajectories. Multidrug resistance, which was observed in some recovered *B. pseudocatenulatum*, may translate into higher survivability during a course of respective antimicrobial treatment, accelerating the recovery of the gut microbiota. Further studies are needed to test this hypothesis, especially given the potential high prevalence of AMR in *Bifidobacterium* in developing country settings.

There are limitations to our study. Since the study design was cross-sectional, we were not able to investigate the microevolution and population structure of *Bifidobacterium* within each participant over time. Our study is limited to genomic profiling, so further work is needed to validate *in silico* predictions related to carbohydrate utilization and the EPS interaction with host cells. In addition, comparison between adult- and child-derived *B. pseudocatenulatum* was preliminary due to the small sample size. These drawbacks notwithstanding, we have a detailed collection of Vietnamese *B. pseudocatenulatum* isolates, from which a candidate for a potential probiotic could be developed. This approach allowed us to tap into the diversity of resident *Bifidobacterium* within an indigenous population, for whom the probiotics or microbiome-targeted complementary foods will benefit. Certain additional features may be considered when selecting a *B. pseudocatenulatum* candidate for development, including immunomodulatory potential (EPS similarity to other efficacious probiotics), metabolic flexibility (extensive glycosyl hydrolase content), survivability under antimicrobial pressures (multidrug resistance), and contribution to overall gut health (production of acetate) (5). Thorough understanding on *B. pseudocatenulatum*'s metabolic capability allows for rational synbiotic design, which optimizes the bacterial survival and colonization in the gut. Regarding AMR, strains with resistance to metronidazole (epigenetic regulation) and/or ciprofloxacin (QRDR mutations) are more fitting, as these resistance determinants are not transferrable to other members of the gut microbiota.

Our study represents the primary step in identifying and characterizing beneficial microbiota members that may represent potential bespoke probiotics from a developing country. This approach could be applied to develop microbiome-mediated therapeutics for other conditions in alternative locations, which will tap into the diversity and functionality held within the gut microbiota of those residing in LMICs, an area which is currently underexplored.

## MATERIALS AND METHODS

**Study design and sample collection.** Samples in this study originated from a cross-sectional study (from May to June 2017) to investigate the gut microbiomes of a healthy Vietnamese population. The study recruited healthy Vietnamese adults aged 18 to 60 who were at the time employed at the Oxford University Clinical Research Unit (OUCRU), Ho Chi Minh City, Vietnam and who were a parent or guardian of a child aged 9 to 60 months whom they also consented to be enrolled in this study. Written informed consent was obtained from the adult participants and from the parent/guardian on behalf of child participants. Participants who had used antimicrobials in the 3 months prior to recruitment, or those who had or were recovering from chronic intestinal disease, chronic autoimmune disease, or allergies, were excluded from the study. Adults who experienced gastrointestinal infections in the last 6 months were also excluded. Based on these exclusion criteria, eligibility for study participation was self-assessed by the study participants. Recruitment was coordinated by a sample manager who ensured participant and specimen anonymity to other study staff. The study eventually recruited 21 adult and 21 child participants. Ethical approval for this study was obtained from the Oxford Tropical Research Ethics Committee (OxTREC, ID: 505-17).

Stool samples were collected from participants using noninvasive procedures. Briefly, participants were provided with a Protocult collection device (Ability Building Center, USA), including a transport container and a Ziploc bag. Specimens were labeled (with participant initials and date of collection) by the participants or their parent/guardian and stored in the freezer until delivery to OUCRU laboratories.

**Shotgun metagenomic sequencing and analysis.** Total DNA extraction was performed on freshly collected stool samples (*n* = 42) using the FastDNA spin kit for soil (MP Biomedicals, USA) following the

manufacturer's procedures. These include a bead-beating step on a Precellys 24 homogenizer (Bertin Instruments, France). All DNA samples were then shipped to the Wellcome Trust Sanger Institute (WTSI, Hinxton, United Kingdom) for shotgun metagenomic sequencing on the Illumina HiSeq2000 platform. All output sequencing libraries were subjected to and passed the quality check on the WTSI pipeline. Taxonomic profiling was performed using the read-based Kraken approach (63) on a curated database of human gut microbial genomes, which include representative genomes (to species level) from the RefSeq database and ones sequenced from the collection in the Lawley lab (WTSI) (64).

**Bifidobacterium culture and identification.** Samples with reads attributed to *Bifidobacterium pseudocatenulatum* above 0.1% of total sequenced reads (7 adults, 9 children) were subjected to *Bifidobacterium* anaerobic culturing using a Whitley A35 anaerobic workstation (Don Whitley Scientific, United Kingdom) containing 5% $CO_2$, 10% hydrogen, and 85% nitrogen gas. Briefly, fecal samples were homogenized in phosphate-buffered saline (PBS; 0.1 g stool per ml of PBS), and several 10-fold dilutions ($10^{-4}$ to $10^{-7}$) were plated onto de Man Rogosa and Sharpe (MRS) media (BD Difco, USA) supplemented with 50 mg/liter of mupirocin (PanReac AppliChem, Germany) and L-cysteine HCl (Sigma-Aldrich, Germany) (14). Plates were incubated at 37°C for 48 h, and ~10 colonies were randomly picked from each fecal sample in a manner to maximize the number of different colony morphologies. Each colony was then restreaked on new MRS plates to confirm purity. For each of these individual bacterial isolates, taxonomic identities were confirmed on the matrix-assisted laser desorption/ionization time of flight mass spectrometer (MALDI-TOF, Bruker). In addition, each isolate was subjected to full-length 16S rRNA gene PCR and capillary sequencing, using the primer pair 7F (5'-AGAGTTTGATYMTGGCTCAG-3') and 1510R (5'-ACGGYTACCTTGTTACGACTT-3') (64). *Bifidobacterium* species identity was confirmed by comparing the output sequence with the NCBI 16S rRNA gene database. In total, 185 isolates were confirmed as *Bifidobacterium* species, of which 49 were *B. pseudocatenulatum* (as identified by both methods).

**Whole-genome sequencing, pangenome analysis, and phylogenetic inference of Bifidobacterium pseudocatenulatum.** Forty-nine confirmed *B. pseudocatenulatum* isolates were subjected to DNA extraction using the Wizard genomic extraction kit (Promega, USA). For whole-genome sequencing (WGS), one nanogram of extracted DNA from each isolate was input into the Nextera XT library preparation kit to create the sequencing library, as per the manufacturer's instruction. The normalized libraries were pooled and then sequenced on an Illumina MiSeq platform to generate 250-bp paired-end reads.

The sequencing quality of each read pair was checked using FASTQC (65), and Trimmomatic v0.36 was used to remove sequencing adapters and low-quality reads (paired-end option, SLIDINGWINDOW:10:20, MINLEN:50) (66). For each trimmed read set, *de novo* genome assembly was constructed separately using SPAdes v3.12.0, with the error correction option and default parameters (67). The median coverage of the assemblies was 56× (range: 26 to 140), with the median number of contigs of 57 (range: 22 to 103) and the median $N_{50}$ of 362,253 bp (range: 135,377 to 1,145,368), showing that the assemblies were of good quality for downstream analyses. Annotation for each assembly was performed using Prokka v1.13, using input of other well-annotated *Bifidobacterium* sequences as references (68). The pangenome of 49 sequenced *B. pseudocatenulatum* genomes, together with public references of the species (DSM20438, CECT_7765, and five assembled genomes from a microevolution study in China [17]), was determined using panX with default settings (69). In brief, panX reconstructs individual gene trees and uses these in an adaptive postprocessing step to scale the thresholds relative to the core-genome diversity instead of relying on a specific single nucleotide identity cutoff. The resulting core genome (1,116 single-copy genes, 137,285 SNPs) was input into RAxML v8.2.4 to construct a maximum likelihood phylogeny of all 56 queried genomes, under the GTRGAMMA model with 500 rapid bootstrap replicates (70). This phylogeny delineates the separation of two lineages, major ($n = 52$) and minor ($n = 4$).

To identify accurately the taxonomic identity of the four isolates belonging to the minor lineage (C01_D5, C01_H5, C01_C5, and C02_A8), we used panX and RAxML as described above to infer the core-genome phylogeny of these four isolates together with several *Bifidobacterium* references. These include *B. adolescentis* 15703, *B. pseudocatenulatum* (DSM20438, 12), *B. catenulatum* (MC1, BCJG468, 1899B, DSM16992, HGUT, BIOMLA1, BIOMLA2, JG), and *B. kashiwanohense* (JCM15439, APCKJ1, PV20-2). For the remaining 52 strains, we mapped each read set to the reference DSM20438 using BWA-MEM with default parameters, and SNPs were detected and filtered using SAMtools v1.3 and bcftools v1.2 (71). PICARD was used to remove duplicate reads, and GATK was employed for indel realignment, as recommended previously (72). Low-quality SNPs were removed if they matched any of these criteria: consensus quality of <50, mapping quality of <30, read depth of <4, and ratio of SNPs to reads at a position of <90%. Bedtools v2.24.0 was used to summarize the mapping coverage at each position in reference 73. A pseudosequence (with length equal to that of the mapping reference) was created for each isolate to integrate the filtered SNPs, region of low mapping coverage, and invariant sites, using the vcf2fa python script (–min_cov = 4, https://github.com/brevans/vcf2fa). Together with the mapping reference, pseudosequences were concatenated to create an alignment, which was then input into ClonalFrameML (branch extension model; kappa = 9.305, emsim = 100, embranch dispersion = 0.1) to remove regions affected by recombination (74). This created an SNP alignment of 10,716 bp, which served as input for RAxML to infer a maximum likelihood phylogeny of 45 *sensu stricto B. pseudocatenulatum* isolates under the GTRGAMMA model with 500 rapid bootstrap replicates (5 iterations). In addition, seven isolates belonging to the C04_cluster were subjected to mapping (to DSM20438) and SNP calling, using the aforementioned parameters. The resulting alignment was input into Gubbins to remove regions of recombination (75), followed by maximum likelihood reconstruction using RAxML.

In order to place the Vietnamese *B. pseudocatenulatum* in a global phylogenetic context, we gathered a collection of global *B. pseudocatenulatum* assembled genomes from the NCBI database (accessed on 5 July 2021), excluding those assembled from shotgun metagenomic data set or with no geographic

attributes. Sixty-eight *B. pseudocatenulatum* genomes were included for downstream analyses, which were isolated from China ($n = 42$), Italy ($n = 1$), Japan ($n = 6$), South Korea ($n = 1$), Spain ($n = 1$), United Kingdom ($n = 5$), and United States ($n = 12$). These were queried with checkm to ensure all had contamination rate of <1.5%, and FastANI was used to confirm that all isolates belonged to *B. pseudocatenulatum* (>97% average nucleotide identity to the reference genome DSM20438) (76). A pangenome was constructed for 45 Vietnamese *B. pseudocatenulatum* isolates in conjunction with all global genomes, using Roary (77). The resulted core-genome alignment was input into ClonalFrameML (branch extension model; kappa = 7.6, emsim = 100, embranch dispersion = 0.1) to remove regions pertaining to recombination. The filtered output alignment (10,595 SNPs) was used for phylogenetic reconstruction by RAxML, under the GTRGAMMA model and with 100 rapid bootstrapping (5 iterations).

**Determination of carbohydrate-active enzymes and antimicrobial resistance determinants.** Representatives from each gene family ($n = 4,333$), as identified in the pangenome by panX as described above, were input into the dbCAN2 metaserver (http://bcb.unl.edu/dbCAN2/blast.php) to annotate genes involved in carbohydrate utilization (78). These carbohydrate-active enzymes (CAZymes) include glycosyl hydrolases (GH), glycosyl transferase (GT), glycosyl lyase, and esterase. A candidate gene was considered a CAZyme if the dbCAN2 output returned any positive hits from the three detection algorithms: HHMER, Hotpep, and DIAMOND. In addition, genomes were screened by ARIBA against a curated resistance determinant database (ResFinder) to detect the presence of acquired resistance genes (–nucmer_min_id = 95, –nucmer_min_len = 50) using input sequencing reads (79, 80). Presence of acquired AMR genes was further detected in compiled global isolates by Abricate (https://github.com/tseemann/abricate), also using the ResFinder database. QRDR mutations were screened manually by aligning the *gyrA*, *gyrB*, *parC*, and *parE* homologs of sequenced genomes, which were retrieved from the constructed pangenome analysis.

**Antimicrobial susceptibility testing of *Bifidobacterium*.** For each PC, we selected one representative isolate for antimicrobial susceptibility testing, except for the case of C16_cluster_1, in which two isolates were included because they showed different genetic composition in the EPS biosynthesis cluster (17 *B. pseudocatenulatum* isolates and 2 *B. catenulatum* isolates). Four control strains were also included: *B. pseudocatenulatum* DSM20438, *B. longum* subsp. *longum* NCIMB 8809, *Staphylococcus aureus* ATCC 25923, and *S. aureus* ATCC 29213. Strains were maintained in brain heart infusion (BHI) with 20% glycerol at −80°C prior to resuscitation on MRS and Luria-Bertani agar (Oxoid, UK) for *Bifidobacterium* and *S. aureus*, respectively. LSM-cysteine formulation (90% Iso-Sensitest broth [Oxoid, UK] and 10% MRS broth, supplemented with 0.3g/liter L-cysteine HCl, with pH adjusted to 6.85 ± 0.1) was chosen for antimicrobial susceptibility testing of *Bifidobacterium*, as recommended previously (81). Muller-Hinton (MH) medium was chosen for testing of *S. aureus*. Prior to testing, strains were precultured on LSM-cysteine agar (48 h for *Bifidobacterium*) or MH agar (24 h for *S. aureus*) under the specified incubation conditions.

For the disk diffusion method, inocula were prepared by suspending *Bifidobacterium* colonies from LSM-cysteine plates into 5 ml of 0.85% NaCl solution (adjusted to McFarland standard 1), which were then spread onto LSM-cysteine plates (82). Subsequently, antimicrobial disks (BioMérieux, France), including ceftriaxone (30 $\mu$g), ciprofloxacin (5 $\mu$g), tetracycline (30 $\mu$g), amoxicillin-clavulanic acid (30 $\mu$g), azithromycin (15 $\mu$g), and metronidazole (5 $\mu$g), were applied. Plates were incubated under anaerobic conditions for 48 h at 37°C, followed by measurement of the diameters of the inhibition zones, including the diameter of the disk (mm). For each isolate, the procedures were repeated five times to evaluate day-to-day reproducibility of the method, with reproducibility defined as percentage of samples within ±4 mm variation in zone diameter (82). For Etests, inocula were prepared as described above, and the resuspended solution was spread onto LSM-cysteine plates. Plates were left to dry for ~15 min, after which Etest strips were applied (ceftriaxone, ciprofloxacin, tetracycline, amoxicillin-clavulanic acid, azithromycin, and metronidazole; BioMérieux, France). The MIC ($\mu$g/ml) was assessed after 48-h incubation, with MIC defined as the value corresponding to the first point on the Etest strip where growth did not occur along the inhibition ellipse.

**Data analysis and visualization.** All data analyses were conducted in R (83) using multiple packages, including ggplot2 and ggtree, for visualization (84, 85). To compare the accessory gene content between child- and adult-derived *B. pseudocatenulatum*, we selected a representative genome from each of the identified phylogenetic clusters (PCs) in the *B. pseudocatenulatum* phylogeny (Fig. 2, $n = 16$). We also included one additional genome for PCs showing high intraclonal variation in the accessory genome (A05_cluster_1, A10_cluster, C04_cluster, C16_cluster_1, C16_cluster_2). This resulted in a set of 21 independent genomes (adult: 10, child: 11). Only gene families (as identified by panX) that are present in 5 to 16 genomes ($n = 606$) were considered for statistical testing (Fisher's exact test) to investigate the differences in genetic composition of the two groups (child versus adult). Due to the limited number of tested genomes, correction for multiple hypothesis testing was not employed, and we reported candidates with *P* value of ≤0.05 as indicating potential differences.

Artemis and Artemis comparison tool (ACT) were utilized to visualize the presence of selected genetic elements in the genomes (86). The EPS biosynthesis cluster was defined as a genomic region flanked by the priming glycosyltransferase *rfbP* or *cpsD* and encompassing several GTs, polysaccharide export *rfbX*, oligosaccharide repeat unit polymerase, tyrosine kinase, and tyrosine phosphatase (47, 51). This region was extracted from targeted *Bifidobacterium* genomes and queried against the NCBI public database using BLASTN to identify the most similar variants. Comparisons between different EPS regions were visualized by Easyfig (87).

**Data availability.** Raw sequence data are available in the NCBI sequence read archive (project PRJNA720750: genomic diversity of *Bifidobacterium pseudocatenulatum* in the Vietnamese population). The accession numbers of genomes used in this study are provided in Table S1.

## SUPPLEMENTAL MATERIAL

Supplemental material is available online only.

**SUPPLEMENTAL FILE 1**, CSV file, 0.01 MB.

**SUPPLEMENTAL FILE 2**, PDF file, 0.6 MB.

## ACKNOWLEDGMENTS

We thank all participants and their parents/guardians for their participation in the study and Magdalena Kujawska for her advice in *Bifidobacterium* culturing. H.C.T. is a Wellcome International Training Fellow (218726/Z/19/Z). L.J.H. is supported by Wellcome Trust Investigator Awards (100974/C/13/Z and 220876/Z/20/Z), the Biotechnology and Biological Sciences Research Council (BBSRC), Institute Strategic Program Gut Microbes and Health (BB/R012490/1), and its constituent projects BBS/E/F/000PR10353 and BBS/E/F/000PR10356. S.B. is a Wellcome Senior Research Fellow (215515/Z/19/Z).

We declare no conflicts of interest.

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
