## [Reviewer comments · Microbiology Spectrum]

Microbiology
Spectrum

Exploring the genomic diversity and antimicrobial susceptibility of *Bifidobacterium pseudocatenulatum* in a Vietnamese population

Hao Chung The, Chau Nguyen Ngoc Minh, Chau Tran Thi Hong, To Nguyen Thi Nguyen, Lindsay Pike, Caroline Zellmer, Trung Pham Duc, Tuan-Anh Tran, Tuyen Ha Thanh, Minh Phan Van, Guy Thwaites, Maia Rabaa, Lindsay Hall, and Stephen Baker

Corresponding Author(s): Hao Chung The, Oxford University Clinical Research Unit

Review Timeline:

Submission Date:

July 26, 2021

Accepted:

August 13, 2021

Editor: Daria Van Tyne

Reviewer(s): The reviewers have opted to remain anonymous.

Transaction Report:

DOI: <https://doi.org/10.1128/Spectrum.00526-21>

August 13, 2021

Dr. Hao Chung The
Oxford University Clinical Research Unit
Molecular Epidemiology
764 Vo Van Kiet
Ward 1, District 5
Ho Chi Minh City
Vietnam

Re: Spectrum00526-21 (Exploring the genomic diversity and antimicrobial susceptibility of *Bifidobacterium pseudocatenulatum* in a Vietnamese population)

Dear Dr. Hao Chung The:

Thank you for revising your manuscript and submitting it to Microbiology Spectrum. I am satisfied with your revisions and I'm happy to inform you that your manuscript has been accepted. I am now forwarding it to the ASM Journals Department for publication. You will be notified when your proofs are ready to be viewed.

Sincerely,

Daria Van Tyne
Editor, Microbiology Spectrum

Journals Department
Fig S1 - S8: Accept
Table S1: Accept